# BRIDGING OFFLINE AND ONLINE REINFORCEMENT LEARNING FOR LLMS

## ABSTRACT

We investigate the effectiveness of reinforcement learning methods for finetuning large language models when transitioning from offline to semi-online to fully online regimes for both verifiable and non-verifiable tasks. Our experiments cover training on verifiable math as well as non-verifiable instruction following with a set of benchmark evaluations for both. Across these settings, we extensively compare online and semi-online Direct Preference Optimization and Group Reward Policy Optimization objectives, and surprisingly find similar performance and convergence between these variants, which all strongly outperform offline methods. We provide a detailed analysis of the training dynamics and hyperparameter selection strategies to achieve optimal results. Finally, we show that multi-tasking with verifiable and non-verifiable rewards jointly yields improved performance across task types simultaneously.

## 1 INTRODUCTION

Large Language Models (LLMs) have demonstrated remarkable capabilities on a wide variety of tasks spanning open ended instruction following to rigid mathematical reasoning (Dubey et al., 2024; Shao et al., 2024). A key ingredient for these capabilities is the "post-training" or alignment stage where a base language model is shaped for specific tasks. During this phase the model is fine-tuned via Reinforcement Learning (RL) to optimize for human preferences or verifiable rewards. The former is suitable for open-ended generations and takes advantage of a reward model during training to reduce reliance on human annotators. The latter is used for math, code, and other less open-ended questions where the correctness of the answer can be directly verified with a boolean score typically by matching against existing labels.

For the optimization method itself, several candidates are commonly considered. When learning from preference labels, Direct Preference Optimization (DPO) (Rafailov et al., 2024) emerged as a powerful algorithm and became a popular choice for open-ended tasks due to its simplistic offline training (Xu et al., 2024a). It can be used with verifiable rewards (Pang et al., 2024) or with reward models (Xu et al., 2023b). DPO can also be used in a semi-online (iterative) fashion (Pang et al., 2024; Yuan et al., 2024). More recently, however, Group Relative Policy Optimization (GRPO) (Shao et al., 2024) has become widely used for fine-tuning LLMs in an online fashion due to its success in training thinking LLMs (Guo et al., 2025). GRPO is based on PPO (Schulman et al., 2017a) which belongs to a class of online RL training methods that try to estimate the gradient of the reward signal.

While recent models have achieved impressive results, the relative importance of various offline to online training approaches and their generalization performance across different tasks remains poorly understood. In this work, we systematically explore the effectiveness of LLM post-training methods in different training setups by bridging the gap between offline and online methods. Specifically, we study offline, semi-online, and online configurations, across both verifiable and non-verifiable tasks, as depicted in Figure 1. By examining the transition from offline to online training, i.e., by altering the speed of periodic model syncing, we aim to understand how these methods can be optimized for improved performance and efficiency on any task. Our investigation focuses on two key aspects: the comparative effectiveness of semi-online or fully online training over offline training and the relative performance of DPO and GRPO objectives across verifiable and non-verifiable tasks.

Based on our experimental results, our contributions are as follows. First, we show that standard DPO lags behind other training regimes significantly, likely due to its offline nature. In contrast, online

Figure 1: **(left):** Visualization of a single training step within our training pipeline, which can be used for any training objective such as DPO or GRPO. Syncing the weights allows rollout responses to be generated from the most recent model. **(right):** Progression from offline to online training, showing when model weight synchronizations occur at different train steps. Offline training only syncs before training starts, whereas online training syncs at every step.

DPO achieves comparable performance to online GRPO, but more surprisingly so does semi-online DPO. We make several recommendations for making such training more stable. The efficiency gains of the semi-online variants opens up an interesting question of whether fully online RL is the only approach for post-training LLMs. Finally, we investigate the performance of joint optimization of verifiable tasks with rule-based rewards and non-verifiable tasks with reward models. We find that this results in improved average results across all tasks compared to the baseline of optimizing only on one objective or the other, as expected, and observe that it improves non-verifiable evaluations compared to only training on non-verifiable rewards.

## 2 LLM ALIGNMENT ALGORITHMS

LLM alignment or post-training is performed after the initial pre-training stage. The de-facto task definition for LLM alignment is an instruction following task where the model input specifies instruction and auxiliary task constraints, and a (typically human-written) response is used as the target. Due to its practical scalability, supervised fine-tuning (SFT) was initially the most common approach to post-train using high-quality instruction following data (Touvron et al., 2023a;b; Zhou et al., 2023). Reinforcement Learning from Human Feedback (RLHF) was proposed before the rise of assistant-like LLMs (Ziegler et al., 2019), and it it was only relatively recently that it was used to outperform SFT methods (Ouyang et al., 2022). This was made possible by instruction following datasets being annotated with a set of responses and human preference labels corresponding to each response, allowing the training of reward models. Initial RLHF models were finetuned using Proximal Policy Optimization (PPO) (Schulman et al., 2017a). More recently, Direct Preference Optimization (Rafailov et al., 2023) and Group Relative Policy Optimization (Shao et al., 2024) have become the gold standard finetuning methods for aligning language models. We detail these methods in the following subsections as they provide the basis for our experiments.

### 2.1 GROUP RELATIVE POLICY OPTIMIZATION (GRPO)

GRPO (Shao et al., 2024) is based on the PPO (Schulman et al., 2017a) algorithm, an on-policy policy-gradient method (Section B.1). While PPO learns from a single sample, which makes it generally applicable, GRPO leverages the fact that we can sample a group of responses $G = \{y^1, \ldots, y^N\}$ for any given prompt $x$. This allows us to approximate a relative advantage of each response by $A(y^i|x) = r(y^i|x) - \sum_{y^j \in G} r(y^j|x)/N$.

$$\mathcal{L}_{\text{GRPO}} = -\mathbb{E}_{G \sim \pi_{\theta_{\text{old}}}} \left[ \sum_{y^i \in G} \sum_t \min \left\{ \frac{\pi_\theta(y_t|x, y_{<t})}{\pi_{\theta_{\text{old}}}(y_t|x, y_{<t})} A(y^i), \text{clip}_\epsilon \left( \frac{\pi_\theta(y_t|x, y_{<t})}{\pi_{\theta_{\text{old}}}(y_t|x, y_{<t})} \right) A(y^i) \right\} \right]. \quad (1)$$

We do not normalize by length like in Shao et al. (2024) because Liu et al. (2025) showed that it can lead to biased optimization, and it is not in the original PPO loss. There is an additional KL term in the loss that we omitted here for brevity.

The main advantage of PPO is that it allows for a small amount of off-policy learning by sampling from an outdated policy $\pi_{\theta_{\text{old}}}$. This enables efficient training by performing multiple updates on the

same batch of generations. The loss uses per-step importance sampling, which is more stable than sequence level importance sampling, and proven to be unbiased (Schulman et al., 2015). The proof relies on the fact that the advantage term is for a single step of the policy

$$A_\pi(y_t) = r(y_t) + V_\pi(y_{t+1}) - V_\pi(y_t). \tag{2}$$

However, the advantage term of GRPO is at the sequence level, so we cannot reuse the same proof and the paper does not provide its own proof (Shao et al., 2024). Therefore, we restrict our experiments to a purely on-policy setup without importance sampling when using the GRPO loss.

## 2.2 DIRECT PREFERENCE OPTIMIZATION (DPO)

DPO (Rafailov et al., 2024) is an offline alignment algorithm that is derived from RLHF (Ziegler et al., 2019; Ouyang et al., 2022) and designed to learn from preference labels $y_c \succ y_r$ where response $y_c$ is deemed better than $y_r$ for prompt $x$. The DPO loss is as follows (see Section B.1 for derivation):

$$\mathcal{L}_{\text{DPO}} = -\log \sigma \left( \beta \log \frac{\pi(y_c|x)}{\pi_{\text{ref}}(y_c|x)} - \beta \log \frac{\pi(y_r|x)}{\pi_{\text{ref}}(y_r|x)} \right). \tag{3}$$

Unlike PPO or GRPO that directly optimize the reward with noisy estimates based on a single sample, DPO optimizes the relation between two samples to match the optimal setup, which can be calculated from data without noise. While this reduced training noise is an advantage, DPO lacks a theoretical guarantee on how a decrease in loss increases the expected reward. Another advantage of DPO, however, is that it does not rely on how the samples are generated, making it appealing for off-policy settings where responses are generated by another model.

## 2.3 SEMI-ONLINE OPTIMIZATION

As described above, GRPO is an on-policy algorithm that requires samples to be generated from the current policy, whereas DPO can learn from off-policy samples (Figure 1). Therefore, the GRPO training pipeline must be online – *i.e.*, the generations and model updates must be synchronous. DPO, on the other hand, was designed for a purely offline setup where we can generate training responses beforehand and train with the DPO loss on these pre-generated responses. However, it is also possible to perform multiple iterations of DPO where one trains on the entire dataset at each iteration, and then generates a new set of responses using the model from the previous iteration. Iterative DPO often offers performance boosts over offline DPO (Xu et al., 2023b; Yuan et al., 2024; Chen et al., 2024b).

In our work we consider a *semi-online* DPO setup where the generation model parameters are synchronized with the training model parameters only periodically, but potentially much more often than in the iterative setting just described. Let $s$ be a number of parameter update steps performed between each synchronization. Decreasing $s$ will make it more online, and eventually become purely online at $s = 1$ when responses are generated using the latest model parameters. In our experiments, we bridge the gap between offline and online training by controlling $s$ to see its effect on downstream performance. The advantage of reducing $s$ lies in computation efficiency where responses can be generated in an embarrassingly parallel way.

While PPO can also be run in a slight off-policy setup thanks to its importance sampling adjustment, it is an inherently on-policy algorithm and uses clipping to limit the importance sampling ratio. In practice PPO is often limited to several update steps before synchronizing the generator with the current model. The GRPO paper (Shao et al., 2024) does not mention if more than one update is performed between synchronizations[1], and most open-source implementations use a pure-online setup. As mentioned in Section 2.1, the off-policy update with GRPO lacks theoretical clarity and is not well studied, so we leave it to future work.

## 3 EXPERIMENTAL SETUP

We study the effectiveness of post-training along three main axes: the training recipe (offline, semi-online, online), algorithm (DPO, GRPO), and tasks (non-verifiable, verifiable).

**Semi-online configurations** We analyze how the update rate impacts training performance and stability. As mentioned before, after every $s$ model weight updates, the generation model is synchronized to match the current model. For both tasks, we compare offline DPO ($s = \infty$), online DPO

---

[1]It mentions "The policy model only has a single update following each exploration stage." which could mean only one update is performed between synchronizations, making it fully online.

and GRPO ($s = 1$), and two semi-online DPO settings that periodically synchronize the generation model ($s \in [5, 10, 100]$). In DPO, we either keep $\pi_{\text{ref}}$ fixed, or update along with the generator model (see Appendix Table 4 for specific implementations).

**Hyperparameter settings** For all tasks, unless otherise noted, we initialize model parameters using `Llama-3.1-8B-Instruct`. During training, we use the default sampling parameters (temperature=1.0, top-p=1.0) to generate exploration rollouts. Other hyper-parameters such as loss configuration, learning rate, gradient clipping, and optimizer settings differ based on the task we train on and are provided in Appendix Table 4.

**Training implementation details** We train all models using the `fairseq2` library (Balioglu et al., 2023), where model inference is performed with the `vllm` library (Kwon et al., 2023). Our main design goal is to create a flexible and modular framework that can easily change policy models, reward models, training algorithms, and datasets. At the same time, we set forth the objective of fast sequence generations for online optimization algorithms such as online DPO and GRPO. We run all experiments using 32 NVIDIA H200 GPUs for training workers and 8 H200s for inference workers (16 for combined tasks). We provide details about the online recipe design in Section C.1.

### 3.1 NON-VERIFIABLE INSTRUCTION FOLLOWING

**Task** Instruction following is an umbrella task that can represent both verifiable and non-verifiable types of questions. Here, we focus on the distribution of problems that users typically ask LLM assistants. We rely on WildChat-1M (Zhao et al., 2024), which is a dataset of 1 million user interactions with ChatGPT. We randomly sample user prompts from the subset of first-turn interactions from the dataset. The prompt template we used for a given instruction is given in Appendix Figure 3.

**Reward** The non-verifiable nature of this task, meaning that there is no (unique) reference answer, requires us to employ a reward model that can estimate the quality of the model response given the user input. We use the open-source LLM-based reward model `Athene-RM-8B` (Frick et al., 2024a), which is experimentally validated as one of the best models to use for preference ranking (Frick et al., 2024b). Athene-RM-8B generates a scalar score for an input-response pair. This allows us to either use the raw response scores as rewards in GRPO, or, in the case of DPO, to rank the responses and create preference pairs of chosen and rejected responses corresponding to the highest and lowest scores, respectively[2].

**Evaluation** For evaluation of the helpfulness and quality of responses, we use AlpacaEval 2.0 (Li et al., 2023b; Dubois et al., 2024) and Arena-Hard (Li et al., 2024c;b) which are robust instruction following benchmarks that have a high correlation with user preferences. We evaluate using two judges: GPT-4-1106 and GPT-4o. We use the decoding temperature $0.6$ and the top-p $0.9$ to generate predictions, which are aligned with the commonly used values of the seed model we use in this work. For training, we use 1,000 WildChat prompts for 1,500 steps. We select the best model checkpoint based on the highest length-normalized Athene-RM-8B rewards on a heldout set of 470 examples: 253 validation examples from Li et al. (2023a) and 218 Evol-Test set examples from Xu et al. (2023a), with prompts that overlap with AlpacaEval 2.0 removed.

### 3.2 VERIFIABLE MATH PROBLEMS

**Verifier** Math problems featuring a reference answer together with the input problem have become a standard in the verifiable training setup (Lambert et al., 2024; Guo et al., 2025). The core component behind such setup is a verifier that can match the predicted answer with the reference one. Some mathematical problems might have multiple written forms of the correct answer e.g., $2/4 = 0.5$ and $2/4 = 1/2$. As such, we use the open-source verifier Math-Verify[3] instead of exact match verification. The template LLM prompt for these tasks is given in Appendix Figure 2. The template asks the model to "reason and give a final answer to the problem" but does explicitly ask for a separate thinking component (Guo et al., 2025) so that it is closer to user instructions from the non-verifiable task.

**Reward** Using the verifier, we obtain binary rewards for each prompt-response pair. For DPO, preference pair selection involves randomly picking the chosen response from the pool of correct predictions, and the rejected response from the pool of incorrect predictions. Prompts that are either

---

[2]While other methods exist to create preference pairs from scalar rewards (Lambert et al., 2024), we choose the best-vs-worst due to its simplicity and stability (Yuan et al., 2024; Xu et al., 2023b; Pace et al., 2024).

[3]https://github.com/huggingface/Math-Verify

Table 1: **Verifiable Task Evaluations**. Test accuracy (std error) for Math500, NuminaMath, and AMC23. Standard error is computed over $N = 50$ random seeds. We find that all semi-online and fully online methods significantly outperform the seed model and offline training, with semi-online DPO, online DPO and GRPO all performing similarly.

| Training method | Math500 | NuminaMath | AMC23 |
|---|---|---|---|
| Seed (`Llama-3.1-8B-Instruct`) | 47.4 (1.6) | 33.9 (0.6) | 23.7 (5.2) |
| Offline DPO ($s = \infty$) | 53.7 (1.6) | 36.4 (0.6) | 28.8 (7.0) |
| Semi-online DPO ($s = 100$) | **58.9** (1.2) | 39.3 (0.4) | **35.1** (5.3) |
| Semi-online DPO ($s = 10$) | 57.2 (1.1) | 39.4 (0.5) | 31.4 (4.3) |
| Online DPO ($s = 1$) | 58.7 (1.2) | **39.6** (0.5) | 32.9 (5.2) |
| GRPO | 58.1 (1.3) | 38.8 (0.5) | 33.6 (5.1) |

too easy or complicated can result in pools where all predictions are correct or incorrect, so we cannot form a valid preference pair. In this case we skip this prompt from the current training step. It is similar in the GRPO loss because all the advantages will be zero.

**Data** We use the NuminaMath dataset (Li et al., 2024a) to collect training prompts and reference answer pairs. During data selection, we filter out problems that require generating a proof, multiple choice questions, and synthetic data, including the Orca math, synthetic AMC, and synthetic math subsets. Proof questions are non-trivial to verify using answer matching, the multiple-choice questions incentivize the model to predict any answer without generating a useful rationale, and the synthetic data may have incorrect answers. After filtering, we end up with a diverse set of 261,440 math problems from which we select 1980 problems each for our held-out validation and test sets.

**Evaluation** We evaluate using Math500 (Hendrycks et al., 2021; Lightman et al., 2023), AMC23, and the NuminaMath test set. We use temperature 0.6 and top-p 0.9 to generate predictions. For each problem we generate $N = 50$ solutions and report the average accuracy as well as the standard error.

### 3.3 COMBINING VERIFIABLE AND NON-VERIFIABLE TASKS

**Skills generalization** While many recent works focus on improving reasoning within specific domains (e.g. verifiable math) (Lambert et al., 2024; Guo et al., 2025), we ultimately want models to perform well on the whole range of tasks. Previous works indicate that using a reward model based on human preferences can lead to reward hacking and poor performance on verifiable tasks (Gao et al., 2023; Guo et al., 2025). We therefore are motivated to study overall performance when only training on one type of reward, and when combining both verifiable and non-verifiable rewards to train a single model. That is, we will use the verifiable rewards for verifiable tasks, and the reward model rewards for non-verifiable tasks. The integration of both types of rewards into a unified training run presents a robust test for the ability of reinforcement optimization generalization. In doing so, we aim to demonstrate two capabilities. First, that we can successfully combine different reward types into a single training run. Second, that the fine-tuned model is both verifiably accurate in definitive math problems, as well as highly coherent and helpful in open ended instruction following problems.

**Data** We consider 3 scenarios: further finetuning a Wildchat Online-DPO-finetuned checkpoint with 100k NuminaMath samples, further finetuning a NuminaMath Online-DPO-finetuned checkpoint with 100k WildChat samples, and finetuning a `Llama-3.1-8B-Instruct` seed model with both WildChat and NuminaMath data. In the last setting, we use 100k NuminaMath prompts and 50k WildChat prompts (since roughly half of the NuminaMath samples are skipped in each batch), and combine samples from both into a single batch. This mixes both verifiable (binary) and non-verifiable (scalar) rewards at each training step. We select checkpoints based on the highest value when averaging length-normalized non-verifiable reward and verifiable reward on their respective validation sets. For all combination training runs, we train for a maximum of 15k steps.

## 4 RESULTS

### 4.1 MAIN RESULTS

**Verifiable math** Table 1 shows math evaluation results for the different training regimes on the NuminaMath training set. The offline DPO training improves performance across all benchmarks compared to the seed model. However, we see substantial gains when training in online, or semi-

Table 2: **Non-Verifiable Task Evaluations**. We show winrate with standard error for length-controlled AlpacaEval, and ArenaHard scores with 95% confidence intervals. Similar to verifiable tasks, both semi-online and online DPO show the best performance, closely followed by GRPO. We show results using two judges: GPT-4-1106 and GPT-4o. While GPT-4o gives overall lower winrates, we see general relative agreement between the two judges.

| Training Method | AlpacaEval LC Winrate | | ArenaHard Score | |
|---|---|---|---|---|
| | GPT-4-1106 Judge | GPT-4o Judge | GPT-4-1106 Judge | GPT-4o Judge |
| Seed (`Llama-3.1-8B-Instruct`) | 27.3 (1.3) | 32.0 (1.55) | 21.3 (-2.2, 1.7) | 27.8 (-2.1, 1.8) |
| Offline DPO ($s = \infty$) | 53.2 (1.5) | 39.4 (1.68) | 38.3 (-2.8, 2.2) | 38.2 (-2.1, 2.9) |
| Semi-online DPO ($s = 10$) | 81.6 (1.0) | 61.1 (1.62) | 59.4 (-1.6, 1.4) | 43.0 (-1.6, 1.6) |
| Semi-online DPO ($s = 5$) | 78.7 (1.2) | 58.5 (1.49) | **60.7** (-1.9, 2.4) | 49.8 (-2.2, 2.2) |
| Online DPO ($s = 1$) | **83.1** (1.0) | **62.1** (1.53) | 60.1 (-1.5, 1.8) | 50.4 (-1.7, 1.9) |
| GRPO | 75.2 (1.2) | 59.1 (1.40) | 55.0 (-1.7, 1.8) | **54.3** (-1.5, 1.6) |

online regimes. We observe several important trends. First, online and semi-online trained models ($s \geq 1$) all outperform the offline DPO model ($s = \infty$) by a wide margin. This highlights the limitation of offline training and the importance of training on responses generated by an updated model. Second, we notice the effectiveness of operating in a semi-online setting with ($s > 1$) for DPO, which performs very similarly to completely online DPO ($s = 1$). This is an important finding indicating that pure online training might not be necessary. We find that online DPO marginally outperforms GRPO. Lastly, we experiment with different numbers of responses in GRPO and report results in Appendix Table 6, where scaling it beyond 8 did not boost performance further.

**Non-verifiable instruction following**  Table 2 compares the performance of different Llama3.1-8b-Instruct models training on WildChat prompts with the `Athene-RM-8B` reward model. We show AlpacaEval-2.0 Length-Controlled (LC) winrates and ArenaHard scores. We observe improvements over the baseline seed model in all training regimes: offline, semi-online, and online. However, again semi-online and online methods significantly outperform the offline DPO results. For example, averaged across both judges, Online DPO results in a 56.6% increase in AlpacaEval LC winrate and 45.6% increase in ArenaHard score compared to the commonly used offline DPO. We also provide results on the `Qwen3-8B` model in Table 5, where we observe similar trends as with Llama.

Similar to the verifiable task setting, online DPO results in slightly higher performance compared to GRPO. Hence both settings emphasize the importance of online and semi-online training methods compared to offline. For semi-online DPO, we test smaller semi-online synchronization step sizes $s = \{5, 10\}$ because 32 steps is already a full data epoch, and we find $s = 100$ to be too unstable with our non-verifiable hyperparameters. We find similar performance between semi-online and online, reiterating the effectiveness of sync step sizes that we observed in the verifiable task. While it is possible that there is some reward hacking with the `Athene-RM-8B` reward model via response length (see Section 4.2), our results demonstrate robust performance on two commonly used instruction following benchmarks that are highly correlated with human preferences and control for common reward hacking such as length and style.

**Combining verifiable and non-verifiable**  Finally, we analyze the effectiveness of training a model with both verifiable and non-verifiable tasks in the training set. Given the strong performance results in the individual verifiable and non-verifiable tasks, and due to computational resource constraints, we only consider online DPO training in this setting. Table 3 shows the results of the combined dataset models compared to training on individual verifiable or non-verifiable tasks. First, we see that the "cross" task performance, training on only verifiable and testing on non-verifiable or vice versa, results in either a decrease in performance or marginal improvement compared to the seed `Llama-3.1-8B-Instruct` model, i.e. there is no transfer. However, we observe significant improvements on non-verifiable tasks when starting from either a WildChat or NuminaMath checkpoint and finetuning on the opposite training set. Notably, even when starting from a checkpoint trained on 1k WildChat and finetuning on 100k NuminaMath samples, we still see gains on non-verifiable evals. We hypothesize that since AlpacaEval and ArenaHard contain "verifiable" prompts such as math and code, it is critical to incorporate some verifiable signal during training as opposed to only using an LLM-based reward. When starting from a NuminaMath checkpoint and finetuning on WildChat, we see a significant drop in math performance as the model starts to optimize for the LLM-based reward.

Table 3: **Combined Verifiable + Non-Verifiable Evaluations**. We first show "cross" task evaluations, when training a model on either the NuminaMath (NM) task only or the WildChat (WC) task only, where we see poor cross task transfer. We then show three separate models trained on both task rewards either starting from a checkpoint trained on the opposite task, or training on both at once. We observe better results with the combined task models across all four datasets than any individual-task model. We show accuracy with standard error for MATH500 and AMC23, winrate with standard error for length-controlled AlpacaEval, and ArenaHard scores with 95% confidence intervals.

| Seed | Training | Dataset | Verifiable | | Non-verifiable (GPT-4o Judge) | |
|------|----------|---------|------------|--|-------------------------------|--|
| | | | **MATH500** | **AMC23** | **AlpacaEval LC** | **ArenaHard** |
| Llama-3.1-70B-Instr. | - | - | 67.2 (1.4) | 46.6 (4.3) | 43.5 (1.6) | 56.4 (-2.3, 2.4) |
| Llama-3.1-8B-Instr. | - | - | 47.4 (1.6) | 23.7 (5.2) | 32.0 (1.6) | 27.8 (-2.1, 1.8) |
| Llama-3.1-8B-Instr. | Online DPO | NM only | **58.7** (1.2) | **32.9** (5.2) | 36.2 (1.6) | 34.9 (-2.2, 2.2) |
| Llama-3.1-8B-Instr. | Online DPO | WC only | 35.0 (1.6) | 15.0 (4.3) | 62.8 (1.5) | 50.4 (-1.7, 1.9) |
| WC-Checkpoint | Online DPO | NM only | 54.7 (1.2) | 30.4 (5.5) | 71.9 (1.5) | **62.3** (-2.1, 2.4) |
| NM-Checkpoint | Online DPO | WC only | 33.3 (1.6) | 13.2 (4.9) | **78.8** (1.6) | 61.2 (-2.0, 2.3) |
| Llama-3.1-8B-Instr. | Online DPO | NM+WC | 57.3 (1.4) | 31.7 (6.0) | 65.6 (1.3) | 57.1 (-2.3, 3.0) |

Lastly, we see performance gains across both verifiable and non-verifiable tasks when starting from the seed model and training on both reward signals. Performance is comparable to training on each task individually, with slight improvements in the non-verifiable evals, demonstrating that not only is it possible to combine rewards during training, it can also help improve performance in certain tasks.

## 4.2 ADDITIONAL EXPERIMENTS AND OBSERVATIONS

**Response length** Although past work has found that both offline and online post-training methods tend to encourage longer answers (Park et al., 2024a; Singhal et al., 2024; Guo et al., 2025), we encounter both length increase and decrease in our training. In the verifiable task, for example, we observe that disabling reference model sync and increasing training speed lead to greater risk of response length collapse and performance degradation (Figure 4). We hypothesize that the bimodal distribution of response lengths (one peak with very short responses, and one with very long responses) is a major contributor to this collapse (Figure 9).

On the other hand, we observe tendencies towards response length increase in the non-verifiable reward experiments. Since we are using an off-the-shelf LLM reward model, the model tends to hack its length bias to maximize rewards (Singhal et al., 2024). Thus, response lengths generally increase in the online or semi-online settings (Figure 7, right). There are several methods to mitigate this: training a reward model for less length bias, incorporating a length penalty in the loss (Wu et al., 2024b; Park et al., 2024b), or selecting checkpoints by normalizing for length. For simplicity across all experiments, we choose the last option and find that this selection method generalizes well.

**Entropy collapse and regularization** We measure the entropy of the next token distribution averaged over all tokens sampled in rollouts in both DPO and GPRO experiments. Figure 5 shows substantial entropy collapse regardless of algorithm in the verifiable task, except for offline DPO. It is possible that offline DPO training is also reducing entropy, but it is not detected here as the measurement is on the rollouts that are not generated from the current model. Non-verifiable tasks, however, exhibit less collapse as training continues (Figure 7, left). This may be due to both the task properties (*i.e.*, gradual improvements, non-binary rewards) and the use of a model-based reward.

We experiment with entropy regularization in the verifiable task to mitigate the entropy collapse in DPO. The average negative entropy of next token distribution is added to the training objective with a configurable coefficient. Empirical results in multiple levels of scale reveal that maintaining stable entropy throughout online training is a non-trivial task, demonstrated in Figure 6, and requires further investigation which we leave for future work.

**Experiments with the loss** Prior work reports benefits of adding an NLL loss over the chosen response $y_c$ to iterative DPO training (Pang et al., 2024; Hong et al., 2024). We experiment with adding the NLL term to our online and semi-online DPO configurations in the verifiable task. We did not observe any benefits after adding the extra term (Figure 10). We explain this observation with the

fact that chosen log probabilities do not substantially decrease during training in our case, which was one of the motivations for adding the NLL term in previous works.

While GRPO trains on all generated responses, DPO only utilizes a pair of responses. In an attempt at improving utilization of the set of generated responses in DPO training, we propose to pair each of the correct responses $\mathcal{Y}_c$ with each the incorrect responses $\mathcal{Y}_r$ in the verifiable task, thus making a *group* of preference pairs. We then average DPO losses off all pairs to compute the *GroupDPO* loss:

$$\mathcal{L}_{\text{GroupDPO}}(x, \mathcal{Y}_c, \mathcal{Y}_r) = \frac{1}{|\mathcal{Y}_c| \cdot |\mathcal{Y}_r|} \sum_{y_c \in \mathcal{Y}_c} \sum_{y_r \in \mathcal{Y}_r} \mathcal{L}_{\text{DPO}}(x, y_c, y_r) \tag{4}$$

We experimented with GroupDPO in the verifiable task setup, as presented in Figure 8, and did not observe substantial performance changes compared to a single preference pair chosen randomly.

Both group DPO and GRPO can learn from all responses in an online manner, but using very different loss functions. This begs the question whether these two losses can be combined. We implemented this loss as $\mathcal{L}_{\text{combined}} = \mathcal{L}_{\text{GroupDPO}}(x, \mathcal{Y}_c, \mathcal{Y}_r) + \alpha \mathcal{L}_{\text{GRPO}}(x, \mathcal{Y}_c \cup \mathcal{Y}_r)$ and train on the verifiable task. We compare the results against semi-DPO and online group DPO in Figure 8 and find no substantial difference in reward or entropy. All relevant hyper-parameters used in additional experiments are provided in Appendix Section C.2.

## 5 DISCUSSION

**Hyperparameter selection** Throughout our experiments, we made several observations about tuning hyperparameters for stable training. In general, we observed frequent instabilities in DPO training that makes the learning suddenly diverge. We found that increasing the Adam epsilon value reduces such collapses and improves the overall stability. The reason for this might be that an epsilon value that is too small forces Adam to make relatively constant updates regardless of the actual gradient value, which could lead to noisy updates when the loss surface is flatter and the gradient is near zero. Increasing epsilon leads to slower convergence, but it can be compensated by increasing the learning rate or gradient clipping.

We experimented with the GRPO loss that has the length normalization, but found it to be less stable. Such normalization will decrease the gradients from longer sequences, which might lead to learning biased towards shorter sequences. We observed a similar trend when adding a length-normalized NLL term to DPO training, where it can boost probabilities of shorter responses more.

**Training Efficiency** One of the key advantages of DPO is its efficiency in requiring only a single pair of responses for each training step on a given prompt. In online DPO, this efficiency remains, with the caveat of having to sample responses and create the pairs at each step. This streamlined approach contrasts with GRPO, which necessitates an entire group of responses, typically more than two, for each prompt. While traditional DPO might be seen as sample-inefficient due to its practice of discarding some responses, the simplicity of needing just two responses per training step can be advantageous. This efficiency reduces GPU memory overhead in the training step, making DPO a more scalable option in compute-constrained settings.

The semi-online configuration brings another advantage. Since the generator model does not need to be synchronized with the trainer during each $s$ step interval, all user prompts from that interval can be annotated with the model's responses asynchronously and in parallel. Speed-up benefits are bound to the technical implementation of these asynchronous annotations, and will scale up as we increase $s$. Such a feature is likely to be particularly attractive in large model post-training where inference is more computationally expensive.

## 6 RELATED WORK

**Reinforcement learning for LLMs** The landmark InstructGPT paper (Ouyang et al., 2022) showed how reinforcement learning from Human feedback (RLHF) (Christiano et al., 2017; Ziegler et al., 2019) can be applied to train instruction following LLMs. This pipeline consisted of the previously standard use of Supervised Fine-Tuning (SFT) training on prompt–response pairs labeled by humans, followed by training a reward model (RM) on human preference rankings over different responses, and finally RL using PPO (Schulman et al., 2017b) with the resulting RM.

**Offline vs iterative vs online training for LLMs** Proposed in 2023, DPO (Rafailov et al., 2024) removes the need for a reward model and directly optimizes for preferred outputs using given pairwise comparisons. This method is offline, and does not depend on producing responses from the model during training. Due to this simplicity and good performance on some tasks, this approach was widely adopted in the community (Tunstall et al., 2023; Mistral AI team, 2023). However further analysis still revealed a gap in performance with online methods like PPO (Xu et al., 2024a; Chen et al., 2024a). Approaches to make DPO semi-online by iterative training, recollecting preferences pairs with the updated model every iteration, showed much stronger results (Xu et al., 2023b; Xiong et al., 2023b; Chen et al., 2024b; Yuan et al., 2024) than standard offline DPO. Completely online variants of DPO were also proposed in Qi et al. (2024); Guo et al. (2024). Xu et al. (2024b) also investigated the tradeoffs between iterative and fully online DPO, finding that semi-online DPO could outperform on-policy DPO when the reference model was synced more frequently. However, their analyses were limited to non-verifiable tasks and relied upon stabilizing online training by setting $\pi_{\mathrm{ref}}$ as an ensemble of multiple sets of LoRA (Hu et al., 2022) weights rather than simply setting $\pi_{\mathrm{ref}}$ to an intermediate checkpoint. Liu et al. (2023) sample preference pairs from the estimated optimal policy, which is closer to online DPO, but is not fully online. Xiong et al. (2023a) demonstrate that RLHF algorithms, including DPO, in general benefit from online exploration. Noukhovitch et al. (2024) introduces asynchronous RLHF, which requires off-policy learning. They observe a decline in RLHF performance with increased off-policy synchronizations, but demonstrates that in general, online DPO is the most robust loss function to off-policy data.

**Non-verifiable vs. verifiable tasks and reasoning models** Much of the work in instruction following training has relied on reward models due to the challenging nature of verifying general tasks including open QA, chat, summarization, and creative writing. With the advent of optimizing reasoning models there has been a renewed interest in verifiable rewards where the task has a known, easily verifiable answer, e.g. short deterministic answers in math problems (Hendrycks et al., 2021; Li et al., 2024a). Pang et al. (2024) showed that Iterative DPO with verifiable rewards could be applied to this setting to improve chain-of-thought reasoning. Lambert et al. (2024) showed that full online RL could be applied with verifiable rewards in a similar setting. DeepSeek-R1 (Guo et al., 2025) applied GRPO (Shao et al., 2024) at scale to this setting, in addition to training on non-verifiable tasks, producing a powerful LLM that can think before answering. Wu et al. (2024a) applied iterative DPO for training such thinking LLMs using non-verifiable tasks.

## 7 CONCLUSION

We explored the effectiveness of various LLM finetuning RL methods across different training paradigms: offline, semi-online, and online, on both verifiable and non-verifiable tasks. Our findings indicate that while offline methods like DPO offer simplicity and efficiency, they often lag in performance compared to their semi-online and online counterparts. We find that fully online DPO and GRPO perform comparably while significantly outperforming offline DPO. Semi-online DPO, which synchronizes the model less frequently than online methods, bridges this gap, nearing the performance levels of fully online methods while allowing the possibility of increased efficiency. The effectiveness of semi-online algorithms makes them viable for agentic applications, where models can continue updating while waiting for long streams of complex rollouts (Silver and Sutton, 2025). This approach builds on asynchronous reinforcement learning principles, demonstrating how offline RL methods like DPO can be adapted to asynchronous settings, what we call semi-online learning.

Additionally, we demonstrate the effectiveness of combining both verifiable and non-verifiable tasks during training. While directly training on only verifiable or non-verifiable tasks yielded limited benefits or even performance decreases on the opposite transfer tasks, starting from a verifiable or non-verifiable trained checkpoint and finetuning on the opposite data type led to significant improvements. Finally, fine-tuning the seed model with a mix of both reward signals achieved improvements on both types of tasks compared to the base model, affirming the benefits of a mixed reward training approach. However, questions remain on the precise best recipe.

Our work provides an exploratory analysis of LLM post-training regimes from offline to offline learning, facilitating further investigation into optimal strategies, particularly around multi-task settings. Due to computational constraints, our experiments are limited to one type of seed LLM model, and future explorations may consider other variants. Future work may explore combining more reward types, including other verifiers, and/or other reward models.

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

# A STATEMENTS

## A.1 REPRODUCIBILITY STATEMENT

We will provide code required to reproduce our results on the benchmark datasets. In the manuscript, we include the type of GPUs that we require to reproduce our experiments. We also provided standard error values for reproducing our results across different seeds.

## A.2 ETHICS STATEMENT

We conform to the ICLR Code of Ethics at https://iclr.cc/public/CodeOfEthics.

## A.3 LLM USAGE STATEMENT

We did not use LLMs for ideation or writing.

# B EXTENDED BACKGROUND

## B.1 LLM ALIGNMENT ALGORITHMS

**Direct Preference Optimization (DPO)** DPO (Rafailov et al., 2024) starts with optimizing the expected sequence-level reward, $r(y)$, with an additional KL term

$$\mathcal{O} = \mathbb{E}_{y \sim \pi} [r(y)] - \beta \text{KL} [\pi(y|x)||\pi_{\text{ref}}(y|x)] \tag{5}$$

where $\pi_{\text{ref}}$ is a reference model (the seed model by default). This objective can be converted into a single KL term (see Rafailov et al. (2023) for proof):

$$\mathcal{O} = -\text{KL} [\pi(y|x)||\pi^*(y|x)], \quad \text{where } \pi^*(y|x) = \pi_{\text{ref}}(y|x)e^{r(y)/\beta}. \tag{6}$$

This shows that the optimal policy for rewards $r(y|x)$ is $\pi^*(y|x)$. Now let us write down the reward that corresponds to the current policy: $r'(y|x) = \beta \log \pi(y|x)/\pi_{\text{ref}}(y|x)$. DPO is designed to learn from preference labels $y_c \succ y_r$ where response $y_c$ is deemed better than $y_r$ for a given prompt $x$. This relation is then converted into rewards by the Bradley-Terry Model

$$p(y_c \succ y_r|x) = \sigma(r'(y_c|x) - r'(y_r|x)) = \sigma \left( \beta \log \frac{\pi(y_c|x)}{\pi_{\text{ref}}(y_c|x)} - \beta \log \frac{\pi(y_r|x)}{\pi_{\text{ref}}(y_r|x)} \right). \tag{7}$$

One can now optimize this using a cross-entropy loss, which gives us the DPO loss

$$\mathcal{L}_{\text{DPO}} = -\log \sigma \left( \beta \log \frac{\pi(y_c|x)}{\pi_{\text{ref}}(y_c|x)} - \beta \log \frac{\pi(y_r|x)}{\pi_{\text{ref}}(y_r|x)} \right). \tag{8}$$

**PPO** PPO (Schulman et al., 2015) is an on-policy policy-gradient method that optimizes

$$\mathcal{L}_{\text{PPO}} = -\mathbb{E}_{y \sim \pi_{\theta_{\text{old}}}} \left[ \sum_t \min \left\{ \frac{\pi_\theta(y_t|x, y_{<t})}{\pi_{\theta_{\text{old}}}(y_t|x, y_{<t})} A_t, \text{clip}_\epsilon \left( \frac{\pi_\theta(y_t|x, y_{<t})}{\pi_{\theta_{\text{old}}}(y_t|x, y_{<t})} \right) A_t \right\} \right]. \tag{9}$$

# C TRAINING DETAILS

## C.1 ONLINE RECIPE TECHNICAL DETAILS

The primary challenge in online training is to enable efficient inference using the latest policy model parameters and optionally the LLM-based reward model. In the framework pipeline, fairseq2's trainer runs as a standard *single program multiple data* (SPMD) run, while generator, reference and reward models run as Ray actors (Moritz et al., 2018) on a standalone Ray cluster. This design allows us to plug in multiple reward models without sacrificing memory capacity of the trainer. Model weight synchronization is done directly between GPU devices using NCCL, and generation communication is done via Ray.

Our training pipeline runs as follows. The process begins with generating policy responses. These responses are then sent to a rewarding unit to compute rewards using either a rule-based system for verifiable tasks or a (LLM-based) reward model. Once the rewards are calculated, a preference or reward batch is composed using the corresponding preference tuning algorithm. This batch is then sent to the preference optimization (DPO) or RL (GRPO) unit to complete the training step.

---

Verifiable task prompt

```
<|start_header_id|>user<|end_header_id|>
Given the following problem, reason and give a final answer to the
problem. Problem:  {PROBLEM}
Your response should end with 'The final answer is $
boxed{[answer]}$.  I hope it is correct.'  where [answer] is the
response to the problem.
<|eot_id|><|start_header_id|>assistant<|end_header_id|>
```

Figure 2: LLM prompt used for verifiable task.

---

Non-verifiable task prompt

```
<|start_header_id|>user<|end_header_id|>
{WILDCHAT INSTRUCTION}
<|eot_id|><|start_header_id|>assistant<|end_header_id|>
```

Figure 3: LLM prompt used for the non-verifiable task.

## C.2 ADDITIONAL EXPERIMENTS HYPERPARAMETERS

In the experiments with adding an NLL loss term we used NLL scale $1.0$. In the experiment with combining GroupDPO and GRPO objectives we tried to scale GRPO loss using scales from a set $\{0.01, 0.001\}$. In the experiments with entropy regularization we tried regularizer scales from a set $\{0.0001, 0.0002, 0.0003, 0.0005\}$.

Table 4: Hyperparameter Settings for Different Tasks.

| Task Type | Task | KL $\beta$ | Learning Rate | Adam $\epsilon$ | Grad Clip | Ref model Sync | Max Len. | Batch Sz. |
|---|---|---|---|---|---|---|---|---|
| Verifiable | Offline DPO | 0.1 | 1e-6 | 1e-4 | 1.0 | No | 2048 | 64 |
| | Semi-Online DPO | 0.1 | 1e-6 | 1e-4 | 1.0 | Yes | 2048 | 64 |
| | Online DPO | 0.1 | 1e-6 | 1e-4 | 1.0 | Yes | 2048 | 64 |
| | GRPO | 0.001 | 1e-6 | 1e-4 | 1.0 | No | 2048 | 64 |
| Non-Verifiable | Offline DPO | 0.01 | 1e-6 | 1e-8 | 0.1 | No | 1024 | 32 |
| | Semi-Online DPO | 0.01 | 1e-6 | 1e-8 | 0.1 | No | 1024 | 32 |
| | Online DPO | 0.01 | 1e-6 | 1e-8 | 0.1 | No | 1024 | 32 |
| | GRPO | 0.001 | 1e-6 | 1e-8 | 0.1 | No | 1024 | 32 |
| Verifiable+Non-Verifiable (NM-Ckpt, WC only) | Online DPO | 0.01 | 1e-6 | 1e-8 | 0.1 | No | 2048 | 32 |
| Verifiable+Non-Verifiable (WC-Ckpt, NM only) | Online DPO | 0.1 | 1e-6 | 1e-4 | 1.0 | Yes | 2048 | 32 |
| Verifiable+Non-Verifiable (Llama-3.1-8B-Instr., NM+WC) | Online DPO | 0.01 | 1e-6 | 1e-5 | 0.1 | No | 2048 | 32 |

## D ADDITIONAL EXPERIMENTAL RESULTS

### D.1 NON-VERIFIABLE INSTRUCTION FOLLOWING WITH QWEN3

In Table 5, we show results on the same non-verifiable instruction following tasks performed with Llama3.1-8B, but this time on Qwen3-8B. We observe similar trends, reinforcing our results.

Table 5: **Non-Verifiable Task Evaluations for Qwen3-8B**. We show winrate with standard error for length-controlled AlpacaEval, and ArenaHard scores with 95% confidence intervals using the GPT-4o judge. Similar to verifiable tasks, both semi-online and online DPO show the best performance, closely followed by GRPO.

| Method | AlpacaEval (GPT-4o judge) | ArenaHard (GPT-4o judge) |
|---|---|---|
| Seed (`Qwen3-8B`) | 64.23 (1.58) | 83.0 (-1.3, 1.5) |
| Offline DPO ($s = \infty$) | 68.15 (1.50) | 85.1 (-1.2, 1.6) |
| Semi-online DPO ($s = 10$) | 77.75 (1.24) | 88.2 (-1.2, 1.6) |
| Semi-online DPO ($s = 1$) | 76.71 (1.34) | 86.7 (-1.3, 1.9) |
| GRPO | 75.35 (1.26) | 90.0 (-1.3, 1.1) |

Table 6: Verifiable task results showing acc (std error). Sampling temperature is set to 0.6, top-p is set to 0.9. Standard error has been computed over $N = 50$ random seeds.

| Training method | Math500 | NuminaMath | AMC23 |
|---|---|---|---|
| Seed (`Llama-3.1-8B-Instruct`) | 47.4 (1.6) | 33.9 (0.6) | 23.7 (5.2) |
| GRPO n=4 | 55.7 (1.4) | 37.7 (0.5) | 30.6 (5.2) |
| GRPO n=8 | 58.1 (1.3) | 38.8 (0.5) | 33.6 (5.1) |
| GRPO n=12 | 57.6 (1.2) | 38.4 (0.6) | 32.2 (5.9) |

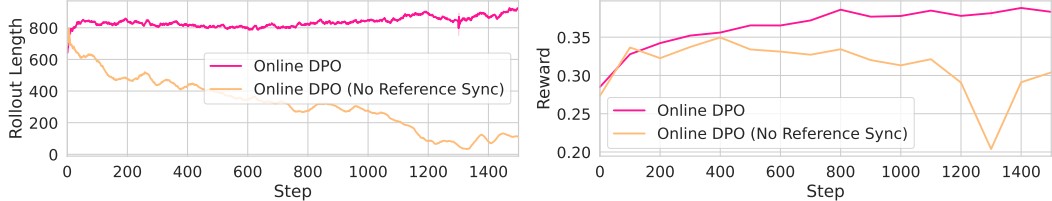

Figure 4: Without syncing the reference model, response lengths of online DPO collapse when trained on verifiable tasks (left). This length collapse is also correlated with lower validation reward (right).

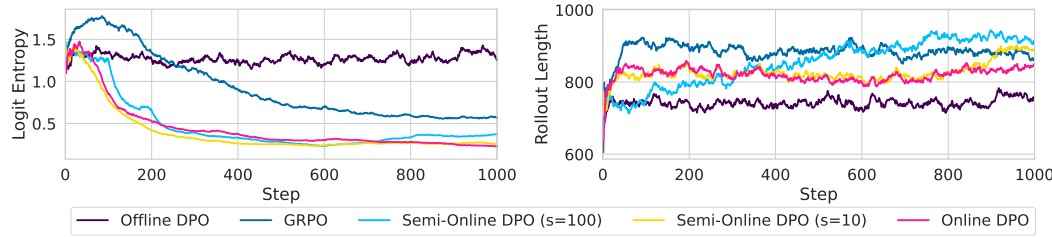

Figure 5: **Logit entropy collapse in iterative and online training on verifiable tasks.** Despite stable average length of rollouts during training (right), the average entropy of the next token distribution (left) decreases significantly during the training in all training regimes except the offline one.

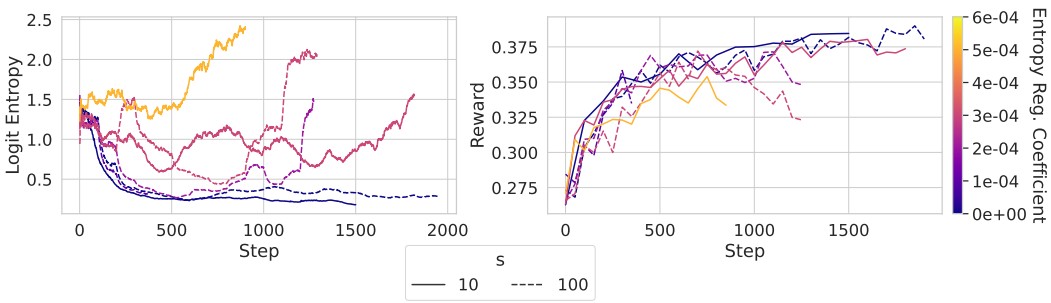

Figure 6: Logit entropy of rollouts and validation rewards of semi-online DPO with (coefficient > 0) and without (coefficient = 0) entropy regularization. Line color indicates strength of the regularization and line style indicates sync intervals.

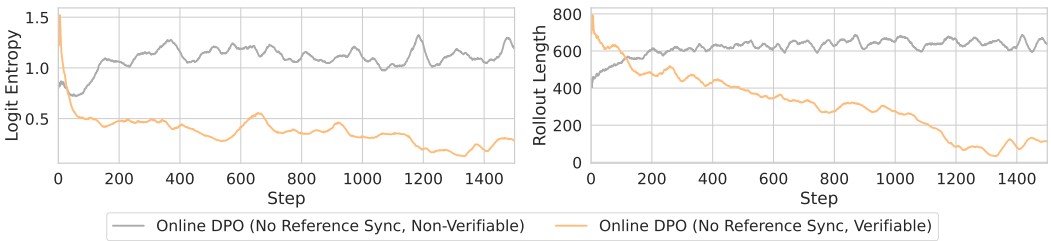

Figure 7: The logit entropy of online DPO trained without reference model synchronization is more likely to collapse when trained on verifiable tasks than on non-verifiable tasks.

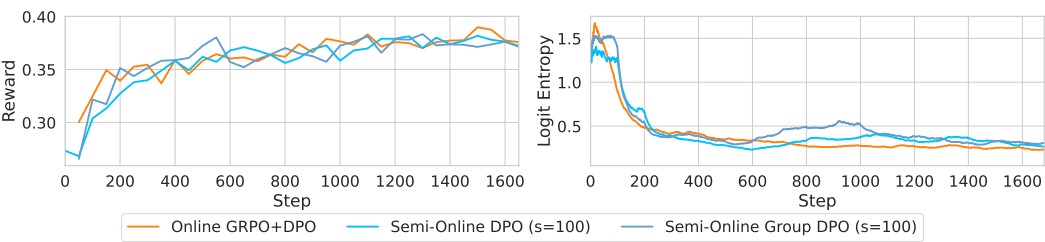

Figure 8: Validation reward and logit entropy of Group DPO, and combining GRPO and DPO compared against semi-online DPO.

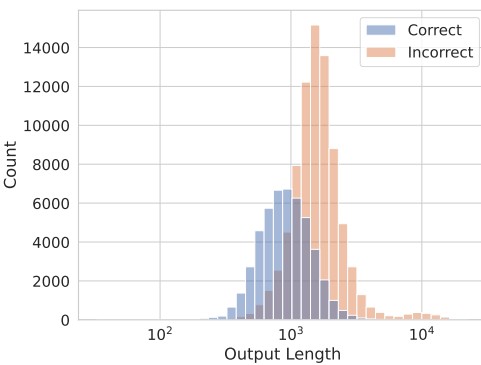

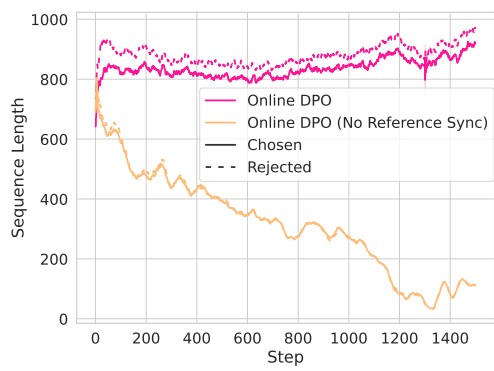

(a) Output lengths of the first checkpoint of the collapsed online DPO run (with no reference model sync) on all math benchmarks.

(b) Lengths of the chosen versus rejected sequences in the training data for the non-collapsed (with reference model sync) and collapsed (without reference model sync) online DPO runs.

Figure 9: At the beginning of online DPO training, the model's shorter responses are more likely to be correct than longer responses (left). If training destabilizes (*e.g.*, due to lack of reference model sync), the model reward hacks by producing excessively short sequences (right). However, if training remains stable, the model learns to gradually increase response length over time.

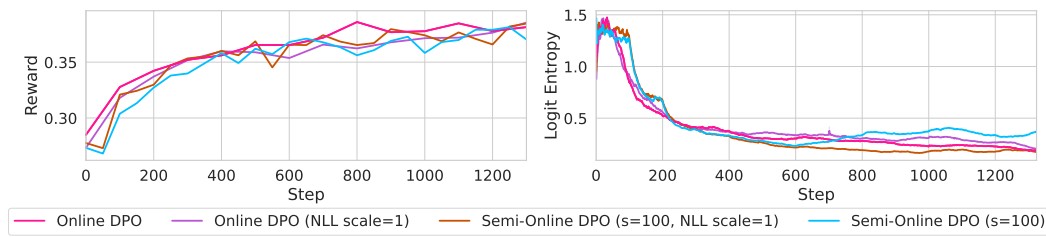

Figure 10: A comparison of online and semi-online DPO with and without an NLL term. Adding an NLL term does not provide benefits for either validation reward nor entropy in these settings.