# OpenReview forum: "Bridging Offline and Online Reinforcement Learning for LLMs"
_ICLR.cc/2026/Conference — Submitted to ICLR 2026_

### Official Review · Reviewer_qMXk · 2025-10-26

**Soundness:** 3
**Presentation:** 3
**Contribution:** 2
**Rating:** 2
**Confidence:** 4

**Summary:**

The authors report the results of on an experimental testbed for LLM training using online and semi-online Direct Preference Optimization and GRPO in math and instruction following tasks. They find similar performance and convergence between these methods.

**Strengths:**

1. DPO performance strongly depends on the coverage and diversity of preference pairs. Thus, an online version of DPO is less susceptible to limited data coverage or out-of-distribution queries.

2. The paper report the results of a comprehensive comparison between DPO (offline and online) and GRPO in math and instruction following tasks. The experiments include variations of DPO that range from fully offline, semi-online (periodic parameter optimization) and fully online.

**Weaknesses:**

1. There is very little novelty in this paper and the conclusions are not surprising: online methods react to changes in distribution and should outperform offline implementation.

2. There are several limitations to DPO the authors fail to address. GRPO or PPO methods can assign rewards to specific parts (tokens) of the generation. In contrast, DPO lacks token-level or step-level credit assignment which are needed in multi-turn or long-horizon tasks. the conclusions of the experiments are biased by failing to include such type of tasks in the testbed.

**Questions:**

1. The testbed should include multi-turn or long-horizon tasks.

---

### Official Review · Reviewer_pqWq · 2025-10-27

**Soundness:** 3
**Presentation:** 3
**Contribution:** 2
**Rating:** 4
**Confidence:** 4

**Summary:**

The paper studies how RL methods for LLM post-training perform when moving from offline to semi-online to fully online settings. It compares Direct Preference Optimization (DPO) and Group Relative Policy Optimization (GRPO) on both verifiable and non-verifiable tasks. Results show that semi-online and online methods greatly outperform offline DPO, with semi-online DPO matching online performance while being more efficient. Combining verifiable and non-verifiable rewards improves overall performance.

**Strengths:**

1. Offers a clear and unified view of offline, semi-online, and online RL using one parameter s; brings together results across verifiable and non-verifiable tasks.

2. Writing is clear and structured; methods and results are easy to follow despite some minor presentation issues.

3. Gives useful, practical insights for efficient LLM post-training; shows semi-online DPO can match online RL and that combining rewards improves generalization.

**Weaknesses:**

**Contribution:**

While the paper’s exploration of semi-online and online DPO is valuable, its conceptual novelty is limited. Prior works (e.g., Xu et al., 2023b; Xiong et al., 2023b; Chen et al., 2024b; Yuan et al., 2024; Qi et al., 2024; Guo et al., 2024) have already proposed iterative and fully online variants of DPO and demonstrated that they outperform offline methods. Therefore, the idea of bridging DPO across offline and online regimes is not fundamentally new. Nonetheless, this paper reinforces the empirical foundation of these prior results by systematically unifying the offline–to–online continuum through a single synchronization parameter $s$. The inclusion of both verifiable and non-verifiable tasks broadens the scope slightly, though it does not represent a major conceptual advance. Overall, the contribution is primarily incremental, providing additional empirical validation rather than introducing a new paradigm.

One of the key contributions of this work, or at least as it appears, is the finding that combining verifiable and non-verifiable rewards yields performance comparable to training on each task individually, sometimes slightly better on one task and slightly worse on the other. However, this result seems rather obvious. The authors never explicitly formulate the optimization problem they are implicitly solving, which effectively reduces to a simple sum of two reward functions weighted by indicator variables denoting the task source (i.e., $R = R_1 \cdot 1_{\text{verifiable}} + R_2 \cdot 1_{\text{non-verifiable}}$). Under such a formulation, it is expected that optimizing for the joint objective would improve average performance on both tasks, with only marginal trade-offs. The observation that joint training can even surpass single-task performance likely stems from the larger effective training set, essentially feeding the model more data, rather than from any genuine synergy between reward types. Moreover, the two tasks are not fundamentally dissimilar, and the verifiable task is structured similarly to the non-verifiable one, further reducing their distinction. In essence, this is a different problem setup, and the results can be generalized as optimization over a weighted sum of rewards, governed by the sampling proportion of each task.

**Experiment:**

Results are mostly limited to Llama-3.1-8B (with minimal results on Qwen3-8B in Appendix). It is unclear if the results are transferable to other families of LLMs.

Results are limited to LLMs with 8B parameters. It is unclear whether the results are transferable to much larger (or slightly larger or even smaller) LLMs.

The study of the combination of rewards omits the case of GRPO and semi-online DPO.

Hyperparameter selection lacks transparency: it is unclear how parameters were chosen (grid search, Bayesian optimization, manual tuning, etc.), what ranges or budgets were used, and how sensitive the results are to these choices.

**Theory:**

While not a major concern, as the work is clearly empirical. We still note that the paper lacks theoretical depth and still does not analyze from a theoretical perspective the explanation for semi-online DPO’s relative success.  Even a brief theoretical or intuitive justification would strengthen the contribution.


**Presentation:**

Syncing is mentioned early on in the introduction but not clearly defined or explained.

Figure 1 shown early on in the paper, and referred in the early paragraphs of the introduction, but it includes several details that are unclear and not easy to follow. S, k, T, G are not explained. The title of the figure should be more comprehensive.

Line 77 repeats the same information from line 76, so it is kind of redundant.

In equation 1, the clip_epsilon is not defined, while it can be retrieved from previous works, the paper should be self-contained in its terms and definitions.

The authors decide to omit the KL term for brevity, but that can mislead the quick readers, should add to equation (1) the term and refer to its full definition in a footnote or appendix.

Equation 3, $\pi_{{ref}}$ used but not defined.

Line 145 It can be unnecessary to emphasize with the term “embarrassingly” parallel way

Line 472 The full term of reinforcement learning is used while being defined as RL (already)

**Questions:**

1. Could the authors clarify what is fundamentally new in their semi-online formulation compared to prior iterative or online DPO work Specifically, how does the introduction of the synchronization parameter sss offer new insights or capabilities beyond existing studies?

2. For the joint optimization over verifiable and non-verifiable rewards, was any weighting or sampling strategy used to balance the two tasks, and did the authors experiment with different ratios to test sensitivity?

3. How sensitive are the main results, especially for semi-online DPO, to synchronization frequency (s) and other key hyperparameters?

4. Have the authors explored whether the observed trends (e.g., semi-online ≈ online performance) hold for larger or different model architectures?

5. Why were GRPO and semi-online DPO not evaluated in the combined reward experiments, and how do the authors expect these variants to perform relative to the tested online DPO model?

---

### Official Review · Reviewer_hx5d · 2025-10-29

**Soundness:** 3
**Presentation:** 3
**Contribution:** 3
**Rating:** 8
**Confidence:** 3

**Summary:**

This paper investigates the difference between offline, semi-offline, and online RL for LLM. It focuses on both verifiable and non-verifiable tasks. It compares two popular algorithms for policy optimization, which are DPO and GRPO. The results show that semi-online and online variants of DPO can perform comparably to fully online GRPO, offering a more training-efficient alternative. The study also explores the effects of mixing verifiable and non-verifiable domains.

**Strengths:**

1. It is a timely and important investigation. The topic directly addresses a practical compute bottleneck for the online GRPO, and it shows trade offs compared to other approaches. The strong performance gain of semi-offline and online DPO shows its viability.

2. It is original. It provides a thorough comparison of the DPO and GRPO which are two main RL methods for LLM post-training, and it provides a new perspective on semi-online DPO as a viable alternative to GRPO.

3. Ablation experiments are comprehensive. It includes entropy and length analysis. It also evaluates the group DPO loss performance.  In addition it evaluates mixed domain training order effects.

4. The findings are clearly communicated, and results are reproducible.

**Weaknesses:**

1. The semi-online step sizes differ across verifiable (s = 10, 100) and non-verifiable (s = 5, 10) settings without clear justification.

2. The study relies on a single model family. It would be much more helpful to show the Qwen model family performance on the verifiable tasks. Right now there are only Qwen results on the non-verifiable evaluations, which makes it difficult to compare to other literature.

3. The claimed training efficiency of semi-online DPO remains qualitative. Quantitative efficiency data and memory would significantly improve credibility, such as wall-clock time, FLOPs, training memory usage, etc.

**Questions:**

See above in Weaknesses

---

### Official Review · Reviewer_GrRG · 2025-11-02

**Soundness:** 3
**Presentation:** 3
**Contribution:** 2
**Rating:** 2
**Confidence:** 3

**Summary:**

The paper provides an evaluation of how off-policy can training go for DPO in various datasets analyzing fully offline, semi-offline and fully online. The paper provides comprehensive evaluations for DPO under this regime.

**Strengths:**

The paper provides an interesting premise where the main point of the study is to see how off-policy one can go before incurring a loss. The y find that one can go somewhat but not fully off-policy which makes sense.

They also evaluate the effectiveness of using verifiable and -non verifiable rewards.

**Weaknesses:**

The paper's premise is good but I think lacks a bit of substance.

The paper reports a single model on three benchmarks and a single seed. I would like to see multiple seeds and another model.

LLM judge as the only evaluation method is in my opinion not enough, I would like to see (even if small) a human study to evaluate the methods.

As far as I can tell the anchor for evaluation is always the base model, it would be nice to see ELO rating to see how the models compare against each other.

This sort of evaluation with both PPO and GRPO would be nice only DPO was evaluated which I think is insufficient. I understand the argument for using DPO with the non-verifiable rewards, but this evaluation can still be done with PPO RLHF style. This analysis with GRPO with only verifiable rewards would also be nice. I understand the authors say that they leave that for future work, but I think such an evaluation is within the scope of this work.

There are some missing references e.g work taht studies trade-offs between offline and online training:

How to Train Your LLM Web Agent: A Statistical Diagnosis


https://arxiv.org/pdf/2507.04103

Pipeline RL


https://arxiv.org/abs/2509.19128

**Questions:**

See weaknesses.

---

### Meta-Review · Area_Chair_s4Yx · 2026-01-02

**Summary:**

Most reviewers express concerns on the lack of novelty, as offline to online training for LLM has been studied extensively before, especially with DPO and its variants. Reviewers also pointed out the evaluation here is limited, due to using a single model and a single seed. Overall most of the reviewers are negative about the paper. As there is no rebuttal, I would proceed to reject the paper based on the reviewers' suggestions.

**Reviewer Concerns:**

The novelty concern remains, as there is no rebuttal.

**Reviewer Scores:**

There is no rebuttal, so the scores will stay as they were.

---

### Decision · Program_Chairs · 2026-01-26

Reject